# Proteomic Analysis of Heavy Metal-Induced Toxicity Using the Cellular Slime Mould *Dictyostelium discoideum*: Effects of Copper Exposure on Aggregation and Protein Expression

**DOI:** 10.3390/toxics13080665

**Published:** 2025-08-08

**Authors:** Atsuko Itoh, Koji Kurihara, Ryo Shoji

**Affiliations:** Department of Chemical Science and Engineering, National Institute of Technology, Tokyo College, 1220-2 Kunugida, Hachioji 193-0997, Tokyo, Japan; cs07069@yahoo.co.jp (K.K.); shoji@tokyo-ct.ac.jp (R.S.)

**Keywords:** *Dictyostelium discoideum*, copper toxicity, proteomics, bioassay, heavy metal contamination, aggregation delay

## Abstract

The cellular slime mould *Dictyostelium discoideum* is a soil-dwelling eukaryotic organism that undergoes distinctive morphological changes during starvation, making it a promising candidate for bioassay development. In this study, we evaluated the effects of copper (Cu) exposure on the morphological transformation of *D. discoideum* and performed a comparative proteomic analysis. Copper exposure on agar media delayed aggregate formation by 3.5 h compared to the controls. Approximately 280 protein spots were detected using immobilised pH gradient two-dimensional gel electrophoresis followed by silver staining. Three spots disappeared upon exposure to Cu. Based on isoelectric point and molecular weight analyses, the proteins were predicted to be formin-1, a cytoplasmic regulator of adenylyl cyclase (CRAC), and a tetratricopeptide repeat (TPR)-containing protein. Formin-1 and CRAC are involved in aggregation processes. These findings suggest that Cu disrupts aggregation-related protein expression in *D. discoideum* and highlight the potential of *D. discoideum*-based bioassays using proteomic biomarkers for environmental monitoring.

## 1. Introduction

Environmental contamination by heavy metals and organic pollutants remains a significant global issue that threatens both ecosystems and human health [1,2]. Bioassays utilising living organisms provide direct and sensitive methods for assessing environmental toxicity. Traditional bioassays often rely on endpoints such as mortality rates or growth inhibition, which require prolonged observation and often lack mechanistic insights into toxicity [3,4].

Recently, there has been increasing interest in understanding the dynamics of how pollutants affect specific cellular signalling pathways and stress responses [5]. *Dictyostelium discoideum*, a eukaryotic microorganism that inhabits forest soils, exhibits a unique life cycle that transitions from unicellular amoebae to multicellular structures under starvation conditions. This morphological transformation is rapid, highly regulated, and sensitive to environmental stressors, making *D. discoideum* a valuable model for investigating cellular responses to toxic substances [6].

Cu was selected as the first-choice toxicant because, although it is an essential trace element, it exerts broad cellular toxicity at elevated concentrations, including disruption of protein structure and induction of oxidative stress [7]. Previous studies have indicated that *D. discoideum* is sensitive to heavy metals such as mercury and copper (Cu) [8,9,10]. However, most assays employ liquid cultures that differ substantially from the natural soil environment of organisms. In this study, we employed solid agar-based cultures supplemented with Cu to mimic natural conditions. We aimed to establish a rapid and sensitive bioassay based on aggregation delay and identify proteomic biomarkers associated with Cu-induced toxicity.

## 2. Materials and Methods

### 2.1. Chemicals

All chemicals, including Cu standard solution (1000 ppm), mycological peptone, yeast extract, and reagents for electrophoresis, were purchased from recognised suppliers (Wako, Kanto Chemicals, Tokyo, Japan, Becton, Dickinson and Company, San Diego, CA, USA, Nacalai Tesque, Tokyo, Japan, and GE Healthcare).

### 2.2. Strain and Culture Conditions

Seed stock was provided by the Science and Education Centre, Ochanomizu University. The AX-2 strain of *D. discoideum* was cultured axenically in AX-2 medium. *D. discoideum* was statically cultured in 4 mL of AX-2 medium at 22 °C under sterile conditions until the density of the cellular slime mould reached 1–2 × 10^7^ cells/mL. A suspension of *D. discoideum* was added to a petri dish containing 4 mL of X-2 medium, and the cell density was adjusted to 1–4 × 10^4^ cells/mL, followed by static culture at 22 °C.

### 2.3. Aggregation Assay

Cells were collected from the bottom of the dish in AX-2 medium, by pipetting with 2 mL Bonner’s salt solution. The bacterial solution was transferred to a tube, stirred for 5 s using a vortex mixer, and centrifuged at 627× *g* for 1 min. The supernatant was discarded, 2 mL of Bonner’s salt solution was added, and the mixture was stirred for 5 s using a vortex mixer and then centrifuged at 627× *g*. This washing procedure was repeated thrice. After washing, cell density was measured using a haemocytometer (Life Technologies Corporation, Carlsbad, CA, USA) and an upright optical microscope (Shimadzu Corporation). The cell density was adjusted to 3.2 × 10^7^ cells/mL by adding Bonner’s salt solution. *D. discoideum* (6.4 × 10^7^ cells) was seeded on non-nutrient agar medium (1.5% (*w*/*v*) agarose) and cultured at 22 °C under light irradiation. The number of aggregates in an area of 1.4 × 1.0 mm^2^ was recorded and observed under a microscope every 30 min for 24 h using the image analysis software program Motic Images Plus 2.1S (Shimadzu Corporation, Kyoto, Japan). The effect of copper exposure was examined by seeding *D. discoideum* on Cu-supplemented agar medium (1 ppm (*w*/*v*) Cu, 1.5% (*w*/*v*) agarose) and performing the same procedure as that for the non-nutrient agar medium. The number of aggregates within a 1.4 × 1.0 mm^2^ area was counted at 14 and 24 h after inoculation on media with or without copper, and Student’s t-test was used to assess whether the differences were statistically significant.

### 2.4. Protein Extraction and Quantification

The protein concentration of the analytical samples was measured by measuring the absorbance using Qubit^®^ Protein Assay (Life Technologies Corporation, Carlsbad, CA, USA) according to the instructions. The protein concentrations of the analytical samples were calculated from the measured absorbance values.

The cells were harvested at the aggregation stage (10.5 h post-seeding). Proteins were extracted by trichloroacetic acid precipitation, solubilised in a buffer containing urea, thiourea, 3-[(3-cholamidopropyl) dimethylammonio]-1-propanesulfonate (CHAPS), and Triton X-100, and quantified using a Qubit Protein Assay Kit.

### 2.5. Two-Dimensional Gel Electrophoresis (2-DE)

In this study, immobilised pH gradient isoelectric focusing was performed using an IPG (immobilised pH gradient strip dry gel) kit from Nihon Eido Co., Ltd. (Tokyo, Japan). Approximately 50 μg of protein from the analysis sample and swelling solution (8 M urea, 4% (*w*/*v*) CHAPS, 65 mM DTT (dithiothreitol), 0.5% (*w*/*v*) ampholines pH 3.5–10, 0.001% (*w*/*v*) BPB (bromophenol blue)) were added to IPG gel (pH 3–10 liner, 7 cm, GE Healthcare, Bucks, UK), and the mixture was allowed to swell for 16–18 h at 23 °C, after which isoelectric focusing was performed. Electrophoresis was performed at 200 V for 10 min, 400 V for 10 min, 1000 V for 1 h, and 1500 V for 18–22 h. After isoelectric focusing, the gel was shaken for 10 min in 4 mL of sodium dodecyl sulphate-polyacrylamide gel electrophoresis (SDS-PAGE) equilibration buffer (0.0625 M Tris-HCl pH 6.8, 25% (*w*/*v*) glycerol, 2% (*w*/*v*) SDS, 0.01% (*w*/*v*) BPB, and 5% (*v*/*v*) 2-mercaptoethanol). SDS-PAGE was carried out according to the method of Laemmli [11] using a separating gel of 10% (*w*/*v*) polyacrylamide containing 0.1% (*w*/*v*) SDS. The acrylamide gel used for the second dimension of SDS-PAGE had a gel size of 8.5 cm × 8.5 cm × 0.1 cm and consisted of a stacking gel of 1 cm and a separating gel of 7.5 cm. The equilibrated gel was placed on top of an acrylamide gel. After the IPG gel was fixed onto the second-dimension SDS-PAGE gel with 1.5% (*w*/*v*) agarose-containing SDS-PAGE equilibration buffer, the gel was electrophoresed up to 1 cm from the anode end of the gel at a constant current of 11 mA/cm^2^ for the stacking gel and 17 mA/cm^2^ for the separating gel.

After electrophoresis, the gel was shaken in 50% (*v*/*v*) methanol plus 5% (*v*/*v*) acetic acid for 14–18 h and then in 50% (*v*/*v*) methanol for 30 min to fix the proteins, followed by silver staining (Wako, Osaka, Japan).

### 2.6. Image and Data Analysis

The silver-stained gel was scanned using a scanner (GT-X900, EPSON, Tokyo, Japan) and EPSON Scan ver. 3.01, and the behaviour of the proteins displayed on the gel was analysed using the differential display method. The differential display method was used to comparatively analyse differences in the expression levels of RNAs or proteins between different samples using PCR and electrophoresis. The colours of the spots in the electropherograms of the Cu-exposed and unexposed samples were replaced with red and green, respectively, using ImageJ ver. to 1.43, and the expression of spots in the Cu-exposed and unexposed samples was compared by overlaying them. Based on the isoelectric point and molecular weight of the proteins whose expression was lost upon copper exposure, the proteins predicted to be expressed were examined using TagIdent. The search conditions were error tolerance of isoelectric point ±0.1 and error tolerance of molecular weight ±5%, and the organism species was limited to *D. discoideum*.

## 3. Results

### 3.1. Morphological Observations

Figure 1 shows the morphological changes in *D. discoideum* grown on non-nutrient agar at different time points after seeding. On non-nutrient agar, *D. discoideum* formed aggregates by 10.5 h after seeding (Figure 1C, arrow). Figure 2 shows the morphological changes in *D. discoideum* grown on Cu-supplemented agar. Aggregate formation was not observed at 10.5 h after inoculation with the control-derived inoculum (Figure 2C), but became evident at 14 h (Figure 2D). This indicates that cells on copper-supplemented agar formed aggregates only after 14 h, reflecting a 3.5 h delay. The number of aggregates significantly decreased after Cu exposure for 14 and 24 h (*p* < 0.05).

Figure 3 shows the time course of the aggregate number changes on the control and Cu-containing agar. Aggregates were detected 10.5 h after starvation in unexposed cells and 14 h after Cu exposure. Cu exposure delayed aggregate formation by 3.5 h and reduced the number of aggregates. Figure 4 shows a comparison of the aggregate numbers at 14 and 24 h with and without Cu exposure. A significant reduction in *D. discoideum* aggregates was observed between Cu-exposed and unexposed conditions both 14 and 24 h after inoculation.

### 3.2. Proteomic Analysis

Figure 5A,B show representative silver-stained gel images of *D. discoideum* proteins. The samples were treated with trichloroacetic acid (TCA), extracted with a protein-solubilising solution containing urea, and separated by immobilised pH gradient two-dimensional electrophoresis combined with isoelectric focusing electrophoresis (IEF) and SDS-PAGE, followed by silver staining. Approximately 280 protein spots were detected in both the control and Cu-exposed samples.

Figure 5C shows the differential display analysis highlighting the protein spots affected by Cu exposure. Figure 5D–I show enlargements of spots a, b, and c, which disappeared with Cu exposure. The spots in the electrophoretic diagram obtained from *D. discoideum* are green for those exposed to Cu and red for those not exposed, and the spots were overlaid to compare the expression levels. Three spots (designated spots a, b, and c) were missing after Cu exposure. Based on database searches, spot a corresponded to formin-1, spot b corresponded to CRAC, and spot c corresponded to a TPR-containing protein (Table 1).

## 4. Discussion

Copper exposure delays aggregation and reduces the number of aggregates formed by *D*. *discoideum*. This suggests that Cu toxicity interferes with the normal developmental program of *D. discoideum*, specifically the transition from unicellular amoebae to multicellular aggregates. Similar inhibitory effects of Cu on developmental processes have been observed in other model organisms, including *Arabidopsis thaliana* and *Caenorhabditis elegans*, highlighting the conserved nature of metal-induced developmental disruptions across species [12,13]. These findings support the potential utility of *D. discoideum* as a sensitive bioindicator.

Proteomic analysis revealed that the expression of three protein spots disappeared after Cu exposure. Based on isoelectric point and molecular weight analyses, the proteins were predicted to be formin-1, cytoplasmic regulator of adenylyl cyclase (CRAC), and a TPR-containing protein. Both formin-1 and CRAC are essential for aggregation in *D. discoideum*, indicating mechanistic disruption of actin cytoskeleton organisation and cAMP-mediated chemotaxis. Formin-1 is involved in the nucleation and elongation of actin filaments, which are crucial for pseudopodia formation and directional cell movement [14]. Previous studies have demonstrated that formin knockout strains of *D. discoideum* exhibit severely impaired aggregation and chemotactic responses [15], reinforcing our interpretation. Although there have been no reports that formin activity is directly affected by copper or other metal ions, Cu^2+^ has been reported to sever actin filaments and shorten their length, which may indirectly interfere with formin-mediated processes [16].

CRAC is critical for relaying cAMP signals and enabling coordinated cell aggregation during starvation. Inhibition or downregulation of CRAC has been shown to result in defective cAMP wave propagation and failure of aggregation centre formation [17]. To date, however, there have been no reports that CRAC is directly affected by Cu^2+^ or other metal ions. In the gill membranes of *Mytilus galloprovincialis*, micromolar concentrations of Cu^2+^ have been reported to strongly inhibit adenylyl cyclase activity, which is activated by CRAC [18]. Thus, the disappearance of CRAC under Cu stress suggests direct impairment of the signal transduction necessary for aggregation, further explaining the 3.5 h delay observed in our morphological assays.

The loss of a TPR-containing protein also implies broader disruption of protein complex formation or intracellular transport. TPR domains mediate protein–protein interactions involved in signal transduction, transcriptional regulation, and chaperone function [19]. Although the specific role of the identified TPR-containing protein in *D. discoideum* has not been fully elucidated, similar proteins have been implicated in stress response pathways in mammals [20], suggesting that Cu may also interfere with cellular stress responses.

It should be noted that our identification was based on bioinformatic predictions, and the lack of definitive identification of the affected proteins remains a limitation. Therefore, future work should involve LC-MS/MS analysis to conclusively identify these proteins and to further clarify the molecular pathways disrupted by Cu exposure.

Our results demonstrate that Cu-induced proteomic changes correlate with measurable developmental phenotypes in *D. discoideum*. Given the organism’s short developmental cycle, genetic tractability, and conservation of key eukaryotic signalling pathways, these findings strengthen the argument for its use in environmental toxicology, particularly in proteomics-based bioassays. Previous studies have proposed *D. discoideum* as a model for neurotoxicity, oxidative stress, and heavy metal responses [21,22]. Our study adds proteomics evidence to this body of work by highlighting the utility of two-dimensional electrophoresis in detecting environmentally responsive protein biomarkers.

## 5. Conclusions

Our study demonstrates that *D. discoideum* aggregation is a sensitive indicator of Cu toxicity and that proteomic profiling can identify key affected proteins. This approach may serve as a basis for the development of rapid and mechanistic environmental bioassays.

## Figures and Tables

**Figure 1 toxics-13-00665-f001:**
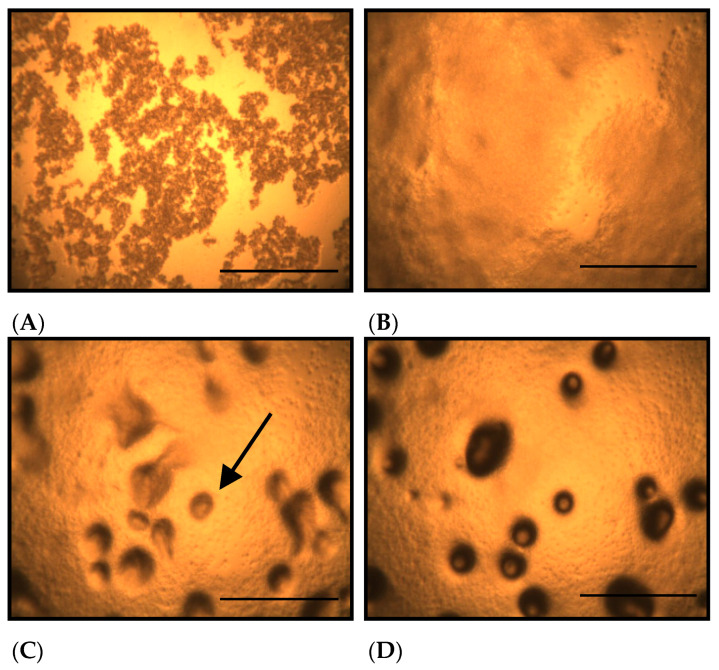
Morphological changes of *D. discoideum* on non-nutrient agar at different time points after seeding. (**A**): Immediately after inoculation, (**B**): 5 h after inoculation, (**C**): 10.5 h after inoculation, (**D**): 14 h after inoculation. The arrows indicate *D. discoideum* aggregates, which have a round outline. No aggregates were observed 5 h after inoculation, but aggregate formation was detected 10.5 h after inoculation. The number of aggregates increased 14 h after inoculation. Scale bars represent 0.5 mm.

**Figure 2 toxics-13-00665-f002:**
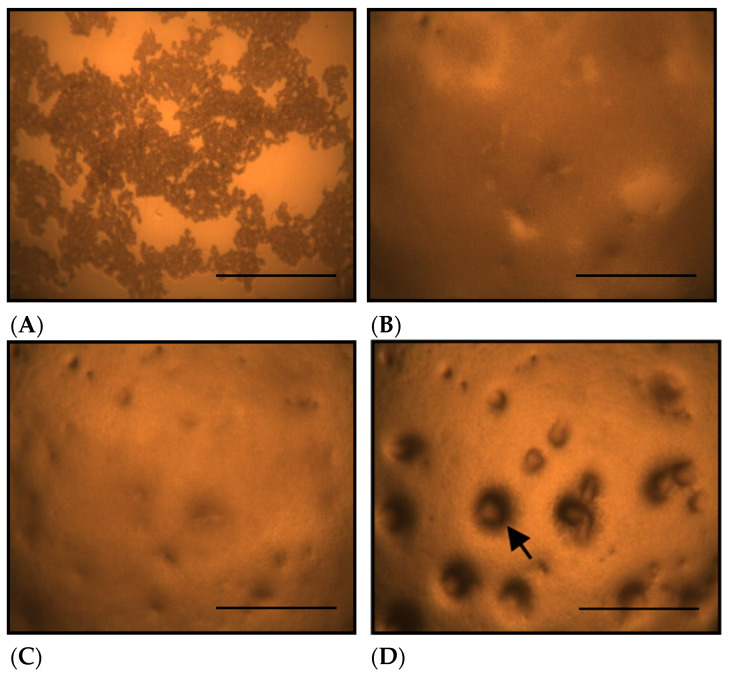
Morphological changes of *D. discoideum* on copper-supplemented agar. (**A**): Immediately after inoculation, (**B**): 5 h after inoculation, (**C**): 10.5 h after inoculation, (**D**): 14 h after inoculation. *D. discoideum* did not form aggregates at 5 or 10.5 hours after inoculation, but aggregate formation was observed at 14 h. The arrow indicates a *D. discoideum* aggregate, which have a round outline. Scale bars represent 0.5 mm.

**Figure 3 toxics-13-00665-f003:**
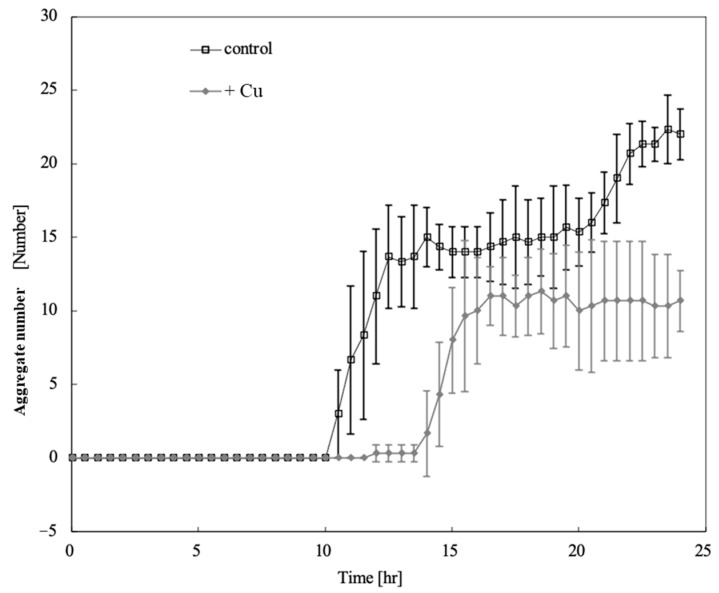
Time course of aggregate number changes on control and Cu-containing agar. Error bars indicate standard deviation. Squares and diamonds indicate the aggregate numbers under control and Cu-exposure conditions, respectively. Aggregates were detected 10.5 h after starvation in unexposed cells and 14 h after Cu exposure. Copper exposure delayed aggregate formation by 3.5 h and reduced the number of aggregates.

**Figure 4 toxics-13-00665-f004:**
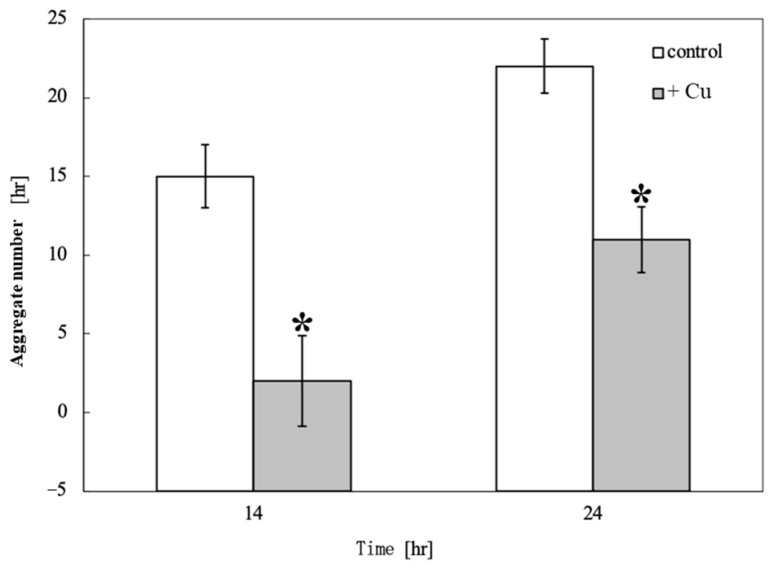
Comparison of aggregate numbers at 14 and 24 h with and without Cu exposure. Asterisks “*” indicate a significant difference in the number of *D. discoideum* aggregates between the Cu-exposed and unexposed conditions, as determined by *t*-test (*p* < 0.05). There was a significant difference in the reduction in the number of *D. discoideum* aggregates between Cu-exposed and unexposed conditions both 14 and 24 h after inoculation.

**Figure 5 toxics-13-00665-f005:**
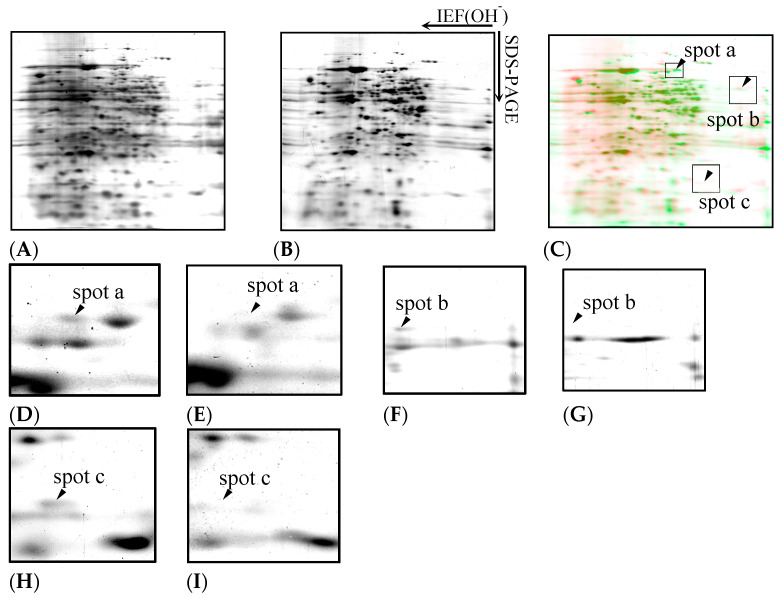
Two-dimensional gel electrophoresis images of *D. discoideum* proteins. The samples were treated with TCA, extracted with a protein-solubilising solution containing urea, separated by immobilised pH gradient two-dimensional electrophoresis combining IEF and SDS-PAGE, and then silver stained. (**A**): Control, (**B**): Cu exposure. Expression of 280 proteins was detected under both copper exposure and non-exposure conditions. The spots in the electrophoretic diagram obtained from *D. discoideum* were coloured green for those exposed to Cu and red for those not exposed, and the spots were overlaid to compare the expression levels (**C**). Cu exposure caused expression of spot a, spot b, and spot c to disappear. Panels (**D**–**I**) show magnified views of spots a to c, as indicated in panel C. Spots a, b, and c detected in the control group (panel **D**,**F**,**H**) are absent under the Cu exposure condition (panel **E**,**G**,**I**).

**Table 1 toxics-13-00665-t001:** Identified proteins corresponding to spots that disappeared under Cu exposure conditions.

Spot	Experiment Value		Protein Name	DictybaseID	ExPasy Data Base		
	pI	Mw (kDa)		DDB	pI	Mw (Da)	Rank
a	6.3	102	formin-1	DDB_G0284519	6.27	106,462	1
			probable ATP-dependent RNA helicase ddx10	DDB_G0284017	6.24	100,359	2
			mental retardation GTPase activating protein homolog 2	DDB_G0270024	6.33	97,411	3
			dynamin-like protein A	DDB_G0268592	6.4	101,404	4
			folliculin interacting protein repeat-containing protein	DDB_G0271996	6.29	105,881	5
			myb-like protein K	DDB_G0276877	6.32	100,912	6
b	8.1	78	cytoplasmic regulator of adenylylcyclase	DDB_G0285161	8.08	78,499	1
			target of Myb protein 1	DDB_G0290163	8.1	75,375	2
			inner no outer 80 complex subunit D	DDB_G0288447	8.07	78,233	3
			cycloartenol synthase	DDB_G0269226	8.19	80,763	4
c	7.5	31	tetratricopeptide repeat-containing protein	DDB_G0285095	7.57	29,748	1

## Data Availability

The data presented in this study are available on request from the corresponding author.

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
