# Peer review of "Proteomic Analysis of Heavy Metal-Induced Toxicity Using the Cellular Slime Mould Dictyostelium discoideum: Effects of Copper Exposure on Aggregation and Protein Expression"

_toxics, 2025, doi:10.3390/toxics13080665_

Round 1
Reviewer 1 Report
Comments and Suggestions for Authors
Review of the manuscript titled "Proteomic Analysis of Heavy Metal-Induced Toxicity Using the Cellular Slime Mould Dictyostelium discoideum: Effects of Copper Exposure on Aggregation and Protein Expression". The work is well-planned in terms of structure, meaning its individual sections are coherent, making it a pleasure to read. The methodological section is also well-planned, making the obtained results credible. The scope of the obtained results is not impressive, but sufficient for a scientific communication. I believe the manuscript will be a good addition to existing knowledge on the presented research topic. Therefore, I believe the work should be accepted for publication in its current form.
Reviewer 2 Report
Comments and Suggestions for Authors
The study’s strength lies in its use of D. discoideum as a model organism to assess heavy metal toxicity, leveraging its well-characterized developmental processes. The proteomic approach to identify differentially expressed proteins (e.g., spots a, b, and c corresponding to formin and CRAC) is promising, and the observed morphological and aggregation changes provide valuable insights.
- The description of Figure 2 (Page 5) as showing “morphological changes” is vague. The authors should specify what changes were observed (e.g., cell shape, size, aggregation patterns) and how they were quantified.
- The identification of spots a and b as formin and CRAC (Page 7) is intriguing, but the manuscript does not explain their biological significance or relevance to copper toxicity. The discussion (if present in the untruncated version) should elaborate on how these proteins relate to aggregation delays or cellular responses to heavy metals.
- The rationale for focusing on copper and the expected proteomic changes should be better justified, potentially by referencing prior studies on heavy metal toxicity in D. discoideum or similar organisms.
Reviewer 3 Report
Comments and Suggestions for Authors
The reviewed article represents a novel method of toxicity testing that demonstrates D. discoideum aggregation can be a sensitive indicator of copper toxicity. This finding complements a way of toxicity testing for other heavy metals or substances. Thus, in my opinion, it is worth publishing.
